# Acute T-Cell-Driven Inflammation Requires the Endoglycosidase Heparanase-1 from Multiple Cell Types

**DOI:** 10.3390/ijms23094625

**Published:** 2022-04-21

**Authors:** Zuopeng Wu, Rebecca A. Sweet, Gerard F. Hoyne, Charmaine J. Simeonovic, Christopher R. Parish

**Affiliations:** 1Genome Sciences and Cancer Division, The John Curtin School of Medical Research, The Australian National University, Canberra, ACT 0200, Australia; zuopeng.wu@health.gov.au (Z.W.); rasweet@gmail.com (R.A.S.); 2School of Health Sciences, The University of Notre Dame Australia, Fremantle, WA 6160, Australia; gerard.hoyne@nd.edu.au; 3Immunology and Infectious Disease Division, The John Curtin School of Medical Research, The Australian National University, Canberra, ACT 0200, Australia; charmaine.simeonovic@anu.edu.au

**Keywords:** heparanase-1 (HPSE-1), extracellular matrix, basement membrane, T cell migration, heparan sulfate destruction, autoimmunity

## Abstract

It has been accepted for decades that T lymphocytes and metastasising tumour cells traverse basement membranes (BM) by deploying a battery of degradative enzymes, particularly proteases. However, since many redundant proteases can solubilise BM it has been difficult to prove that proteases aid cell migration, particularly in vivo. Recent studies also suggest that other mechanisms allow BM passage of cells. To resolve this issue we exploited heparanase-1 (HPSE-1), the only endoglycosidase in mammals that digests heparan sulfate (HS), a major constituent of BM. Initially we examined the effect of HPSE-1 deficiency on a well-characterised adoptive transfer model of T-cell-mediated inflammation. We found that total elimination of HPSE-1 from this system resulted in a drastic reduction in tissue injury and loss of target HS. Subsequent studies showed that the source of HPSE-1 in the transferred T cells was predominantly activated CD4^+^ T cells. Based on bone marrow chimeras, two cellular sources of HPSE-1 were identified in T cell recipients, one being haematopoiesis dependent and the other radiation resistant. Collectively our findings unequivocally demonstrate that an acute T-cell-initiated inflammatory response is HPSE-1 dependent and is reliant on HPSE-1 from at least three different cell types.

## 1. Introduction

One of the remarkable features of the adaptive immune system is the ability of antigen-specific T lymphocytes, whether pathogen- or autoantigen-specific, to traverse extracellular matrices (ECM) and basement membranes (BM) when being recruited from the circulation into inflammatory sites. Although mechanical forces and invadopodia are utilised during development of various multicellular organisms for a similar outcome [1,2], proteases that degrade ECM and BM barriers are regarded as the main mechanism that facilitates lymphocyte migration through ECM and BM barriers to sites of infection [3,4,5]. However, paradoxically, drugs that inhibit proteases have been found to be not as effective at retarding cell migration through ECM and BM as initially anticipated based on in vitro studies, this lack of efficacy being particularly evident in the context of cancer metastasis [6]. Mice genetically deficient in various proteases have also been shown to behave as wild-type [7]. It has been proposed that this paradox is due to the large number of proteases that can potentially degrade ECM and BM and, consequentially, provide marked redundancy for most proteases. An exception to this rule is the endoglycosidase heparanase-1 (HPSE-1), which is the only enzyme of its type that can degrade heparan sulfate (HS), a major constituent of ECM and BM that is responsible for BM integrity and barrier function. Thus HPSE-1 is thought to play a key role in the solubilisation of ECM and BM by infiltrating lymphocytes [5]. Conversely, there is emerging evidence that ECM and BM can be traversed by some cells without involvement of degradative enzymes such as HPSE-1 [2,8].

Since HPSE-1 is the only known endoglycosidase that can degrade HS, using an adoptive transfer model in which transferred T lymphocytes are genetically deficient in this enzyme should be ideal for determining the importance of ECM/BM solubilisation by HPSE-1 in T-cell-mediated inflammatory responses. In this regard, we have taken advantage of the well-known adoptive transfer model of autoimmune Type 1 diabetes in which ovalbumin (OVA)-specific T cells are transferred into recipient mice expressing OVA in insulin-producing beta cells in pancreatic islets [9]. The net result of this model is acute inflammation in the recipient islets which induces islet beta cell destruction and the development of Type 1 diabetes. An additional unique feature of this model is that pancreatic islets contain high levels of HS which is critical for beta cell survival, with loss of islet HS correlating with disease severity. Surprisingly, we found in this model that there are roles for HPSE-1 produced by multiple cell types. Thus, antigen-specific T cells in the CD4^+^ but not CD8^+^ subsets require HPSE-1 in order to induce strong inflammation and end-point disease identified by hyper-glycemia or elevated blood glucose levels. Furthermore, we found a role for host-cell-derived HPSE-1 and this includes both radiation-resistant and haematopoietic-derived bone marrow cells. Based on our findings, this study is the first to show the co-operation of different cell types to provide a specific degradative enzyme, in this case HPSE-1, required for the initiation of an acute, CD4^+^-T-cell-dependent inflammatory response.

## 2. Results

### 2.1. Antigen-Specific T cell Transfer Model for Studying the Role of HS Degradation in T-Cell-Mediated Inflammation

In order to investigate the role of the ECM-degrading enzyme HPSE-1 in T-cell-dependent inflammatory responses, we took advantage of a well-characterised model of inflammation that results in destruction of the insulin-producing Islets of Langerhans in the pancreas, with diabetes induction being a convenient measure of inflammation intensity. In this model, co-transfer of naïve CD8^+^ OT-I and activated CD4^+^ OT-II T cells, each with a specificity for OVA, into RIP-OVA^hi^ mice that express islet-specific OVA antigen, induces rapid diabetes development [9]. In fact, this model mimics that seen normally in vivo where antigen-activated CD4^+^ T cells are required to recruit CD8^+^ T cells into inflammatory sites [10,11].

We have, however, modified this model slightly in order to yield more reproducible results. First, additional OVA_323–339_ peptide was included in the culture media to overcome the lower TCR affinity of the OT-II T cells and improve OT-II T cell activation, viability, and yield. Second, depletion of B cells, NK cells, macrophages, dendritic cells, granulocytes, red blood cells, and other T cell subsets with appropriate monoclonal antibodies and magnetic beads, rather than by complement-mediated lysis, optimised OT-I and OT-II T cell purity to >95% for adoptive transfer. After titration of donor T cell numbers, we found that 2 × 10^6^ naïve OT-I together with 2 × 10^6^ activated OT-II T cells efficiently induced diabetes onset in all the recipient mice at around 10 days post-transfer, whereas the same number of OT-I or OT-II T cells transferred alone did not (Figure 1A,B).

In accordance with hyperglycemia, severe islet inflammation was found in recipients two weeks after receiving both naïve OT-I and activated OT-II T cells (Figure 1C). All islets had various levels of lymphocyte infiltration and 80% of the islets were destroyed. Although no diabetes was found when OT-I or OT-II T cells were transferred alone, activated OT-II T cells induced mild to moderate infiltration in 40% of host islets. By contrast, very little damage was caused by naïve OT-I T cells transferred alone.

Previously we have shown that pancreatic islets contain extraordinarily high levels of HS [12,13], the HS being lost during T1D development in non-obese diabetic (NOD) mice and proposed to protect islets from reactive oxygen species damage [12,13,14,15,16]. Thus, islet HS content represents a direct and sensitive measure of the damaging effects of T-cell-mediated inflammation in the context of autoimmune diabetes. In order to test whether HS content was affected in the islets of the adoptive transfer recipients, we stained pancreas sections with Alcian blue, a histochemical method for detecting HS in tissues [13,17]. Interestingly, the combination of OT-I and OT-II T cells caused significant HS loss (Figure 1D), many islets being devoid of HS and becoming gold in colour, a finding which correlated with the islet inflammatory infiltrates and diabetes development in these recipients. In contrast, we found that transfer of OT-I and OT-II T cells alone led to a partial loss of HS within islets of RIP-OVA^hi^ mice, with activated OT-II T cells showing a stronger capability than naïve OT-I T cells at depleting HS within islets (Figure 1D). This was not due to the difference in the activation status of the OT-I and OT-II T cells being adoptively transferred because circulating OT-I T cells were quickly activated and upregulated CD69 expression as early as one day after adoptive transfer into the hosts (data not shown). Furthermore, consistent with the results above, we found that OT-I and OT-II T cells increased expression of the HS-degrading endoglycosidase HPSE-1 on their cell surface after antigen activation in vitro (Figure 1E), implying that these antigen-specific T cells could use HPSE-1 to initiate the inflammatory response and mediate the HS loss observed in this model.

### 2.2. HPSE-1 Drives T-Cell-Mediated Inflammatory Responses

Given that HPSE-1 is expressed by antigen-activated CD4^+^ and CD8^+^ T cells in this model of inflammation and that HS is degraded in the islets, we next tested the requirement for HPSE-1 in these processes using HPSE-1-deficient mice [18]. Thus, we adoptively transferred HPSE-1 WT or HPSE-1-deficient OT-I with OT-II T cells into two groups of either HPSE-1 WT or HPSE-1-deficient RIP-OVA^hi^ recipients (Figure 2A). WT and HPSE-1-deficient OT-II T cells were prepared separately but stimulated in vitro under the same conditions, purified with the same protocol and shown to have equivalent viability and activation status, based on CD44 expression (data not shown). They were then mixed with equivalent numbers of WT or HPSE-1-deficient naïve OT-1 T cells and transferred into age- and sex-matched recipients. As before, all of the WT recipients receiving WT T cells became diabetic within 2 weeks (Figure 2B). Interestingly, HPSE-1 deficiency alone in either the donor T cells or the recipient hosts delayed diabetes onset by 2–3 days but all animals became diabetic. In contrast, and strikingly, 60% of HPSE-1-deficient hosts receiving HPSE-1-deficient T cells were diabetes free during the same period. Thus, along with our previously published finding that the HPSE-1 inhibitor PI-88 is effective in ameliorating diabetes in NOD mice [13], this study further supports the importance of HPSE-1 as a therapeutic target in disease conditions involving T-cell-mediated inflammatory infiltrates, such as seen in T1D.

To further investigate how HPSE-1 contributes to the inflammatory response in this model, we next examined pancreas histology. As before, in the WT group all of the islets tested had various grades of leukocyte infiltration and most of them were fully destroyed or severely damaged (Figure 2C). Absence of HPSE-1 in either donor T cells or recipient mice decreased the severity of islet inflammation (Figure 2C). Remarkably, in line with the incidence of diabetes described above, when both donor T cells and recipient animals were HPSE-1-deficient a significant proportion of islets (20%) were still intact and a further 30% of islets contained only a mild leukocyte infiltrate. When HPSE-1 expression was restricted solely to either antigen-specific T cells or recipient mice, intra-islet HS was found to be dramatically reduced in both groups, similar to that seen when both donor T cells and recipients were WT (Figure 2D). Thus, HS loss is associated with moderate to severe islet inflammation and resulted in a delay in diabetes onset. By contrast, the overall intra-islet HS content remained relatively normal in the totally HPSE-1-deficient group and was well preserved in most of the islets including many of those with destructive insulitis. Thus, HS loss was prevented by a complete deficiency in HPSE-1 and was accompanied by a dramatic reduction in diabetes incidence.

Although HPSE-1 played a clear role in the outcome of our model, we wanted to understand the events leading up to this outcome. To assess the kinetics of intra-islet HS degradation, lymphocyte infiltration, and inflammatory responses initiated by self-reactive T cells, RIP-OVA^hi^ hosts were examined at different time points prior to diabetes onset. Interestingly, in the presence of HPSE-1 there appeared to be a partial, but not significant, intra-islet HS loss seen at 3 days post-transfer (Figure 2E), despite no obvious histological inflammation (Figure 2F). From day 6 to day 9, there was a sharp and significant decline in islet HS content and, coincidently, inflammation of islets worsened quickly during this period (Figure 2E,F). Of note, HPSE-1 deficiency delayed and dampened HS degradation, along with less severe islet inflammation at these early time points (Figure 2E,F).

### 2.3. HPSE-1 from Antigen-Specific CD4^+^ T Cells Collaborates with Host HPSE-1 from Other Sources to Enable Islet Infiltration

The experiments above tested the role of HPSE-1 when it was either present or absent collectively from both CD4^+^ and CD8^+^ antigen-specific T cells. We next wanted to dissect the separate roles of HPSE-1 in antigen-specific CD4^+^ and CD8^+^ T cells in the process of islet infiltration and diabetes pathogenesis. As described above, OT-II T cells alone caused more dramatic intra-islet HS loss and islet inflammation than OT-I T cells (Figure 1D). The question was whether HPSE-1 in OT-I or OT-II T cells was crucial in diabetogenesis if HPSE-1 was not simultaneously expressed in both OT-I and OT-II cells. To investigate this, we limited HPSE-1 expression to antigen-specific CD4^+^ T cells (OT-II), CD8^+^ T cells (OT-I), or both, with host bystander T cells and other host cells being HPSE-1 deficient (Figure 3A). The results showed that WT OT-II T cells caused 60–70% diabetes incidence, regardless of whether or not the OT-I T cells expressed HPSE-1; by contrast, deficient OT-II T cells only induced diabetes in 20% of recipient mice, even when accompanied by WT OT-I T cells (Figure 3B). These data suggest that HPSE-1 in antigen-specific CD4^+^ T cells plays a pivotal role in the autoimmune attack of beta cells by degrading intra-islet HS and initiating autoimmune and inflammatory responses.

Although HPSE-1 from antigen-specific CD4^+^ T cells is clearly involved in diabetes initiation, it did not attain the diabetes incidence observed in WT controls. This result implied that activated CD4^+^ T cells are not the only source of HPSE-1 contributing to HS degradation and islet infiltration. We next asked if HPSE-1 derived from host hematopoietic cells or radiation-resistant cells contributes to diabetes induction with HPSE-1-deficient T cells. We prepared bone chimeras, using established protocols [9,19,20], to address this issue. We reconstituted HPSE-1 WT and HPSE-1-deficient RIP-OVA^hi^ host mice with either HPSE-1 WT or HPSE-1-deficient BM cells. We then adoptively transferred HPSE-1-deficient OT-I and OT-II T cells and monitored diabetes development (Figure 3D). In this way, HPSE-1 was restricted to either or both of the two distinct host cell populations whereas antigen-specific T cells were HPSE-1-deficient. In chimeras of WT hosts reconstituted with WT BM, the results resembled diabetes induction in WT hosts by WT T cells, i.e., all mice became diabetic (Figure 3D). Similar to the results above, chimeras of deficient hosts and deficient BM were only ~20% diabetic after deficient antigen-specific T cell transfer. Chimeras of HPSE-1 WT hosts with deficient BM or the reverse had virtually comparable and intermediate diabetes incidence. These data provide evidence for both hematopoietic and radiation-resistant cells being sources of HPSE-1 that enable HPSE-1-deficient OT-I and OT-II T cells to induce Type I diabetes.

### 2.4. HPSE-1 Drives T Cell Infiltration of Peripheral Tissues

The kinetics of entry of antigen-specific T cells into islets during T1D pathogenesis is not fully understood. With the advantage of this adoptive T cell transfer model, we tracked the diabetogenic OT-I and OT-II cells by labelling them with two different fluorescent dyes, CTV and CFSE [21], and found their early presence on day 4 in the islets of RIP-OVA^hi^ mice during diabetes development (Figure 4A,B). Furthermore, despite the low numbers, HPSE-1-deficient OT-I and OT-II T cells were found at a significantly reduced frequency in the islets even just 4 days after transfer (Figure 4B). Thus, inhibiting initiation of infiltration by HPSE-1-deficient antigen-specific T cells at early time points is likely to contribute to the decreased severity of the responses at much later time points, e.g., day 14 (see below).

Further tracking of HPSE-1-deficient OT-I and OT-II T cells in the islets of HPSE-1-deficient hosts also revealed reduced numbers compared with their WT counterparts on day 6 post transfer (Figure 4B), although this difference on day 6 was not statistically significant. Interestingly, significantly fewer macrophages were also found in the deficient group compared with the WT group on both day 4 and day 6. These data demonstrate a phenomenon of HPSE-1-dependent intra-islet HS degradation and infiltration of both OT-I and OT-II T cells and macrophages in the early phase of disease initiation in this highly aggressive type 1 diabetes model. An intriguing question that arises from this study is: what T-cell-derived cytokines and chemokines are responsible for the recruitment of macrophages into islets? Although there is a plethora of chemokines that can attract macrophages to sites of inflammation, very few appear to be produced by CD4^+^ T cells. Thus, it seems more likely that CD4^+^ T cells, following activation and proliferation by a specific antigen, release primary cytokines such as IL-1, IFN-γ, and TNF, which induce the activation of other immune cells, such as macrophages, DCs, and monocytes, to produce chemokines. In this regard, pancreatic islets have been shown to produce and to respond to a wide range of chemokines that are presumed to regulate islet inflammation [22].

Finally, we analysed OT-I and OT-II T cells at a late time point, i.e., 14 days. We used the congenic marker CD45.1 [20] to quantify the transferred OT-I and OT-II cells as the frequency of CD8^+^TCRvα2^+^ and CD4^+^TCRvα2^+^ cells, respectively, in the draining pancreatic lymph nodes (PLN) (Figure 4C,D). Our data show significantly reduced frequencies of HPSE-1-deficient OT-I and OT-II T cells in the draining PLN of deficient recipients compared with the other three groups (Figure 4D). By contrast, in spleen and non-draining LNs, similar frequencies of donor T cells were seen in all groups and there were fewer transferred cells in these other tissues compared with those observed in the PLN (data not shown). This suggests that tissue-specific OVA expression drives OT-I and OT-II T cell homing. Thus, absence of a lymphoid circulation was not the mechanism underlying reduced responses in the pancreas of HPSE-1-deficient OT-I and OT-II T cells; instead, functional absence of HPSE-1 during the inflammatory response appears to reduce OT-I and OT-II T cell expansion in the draining LN. Overall, HPSE-1 derived from antigen-specific T cells works in concert with at least two other sources of HPSE-1 within the host to facilitate inflammatory T cell invasion of peripheral tissues.

## 3. Discussion

HPSE-1 is the only known mammalian enzyme that is capable of degrading HS, a glycosaminoglycan that is a major constituent of ECM and BM [5,23]. Although there are a number of HS-degrading enzymes involved in HS disassembly, HPSE-1 is the only one that can act as an endoglycosidase, making it capable of cleaving HS chains at a number of specific internal sites, like a restriction enzyme cleaves DNA. This makes it a unique enzyme that can solubilise HS chains embedded in ECM and release HS-bound growth factors and cytokines as well as aiding passage of cells through tissues by facilitating the solubilisation of ECM and BM [5,24,25]. However, as outlined earlier, there is emerging evidence that cells may be able to pass through ECM and, in particular, formidable BM barriers by mechanisms that do not require degradative enzymes but, for example, employ invadopodia and mechanical forces [1,2].

Based on the above, this study was undertaken to examine whether or not activated T cells require BM-degrading enzymes in order to enter inflammatory sites, with HPSE-1 being the enzyme of choice. This is due to HPSE-1 being the only endoglycosidase identified in mammals that is able to cleave HS, a major constituent of the subendothelial BM in the walls of blood vessels that the T cells must traverse. The other key features of this study were the availability of *Hpse^−/−^* mice [18] and the use of the OVA-specific CD4 OT-II and CD8 OT-I adoptive transfer model of autoimmune diabetes which results in destruction of OVA-expressing insulin-producing pancreatic islet beta-cells, i.e., RIP-OVA^hi^ cells [9]. Our initial studies confirmed the previously published results obtained with this model, diabetes being only observed in RIP-OVA^hi^ mice receiving both naïve OT-I and antigen-activated OT-II T cells (Figure 1B). Destructive islet inflammation also correlated with diabetes induction (Figure 1C). We have previously reported that islet beta cells express very high levels of intracellular HS, which we believe maintains islet viability by acting as a sink for reactive oxygen species [12,13,14,15,16]. Consistent with this view, it was found that there was a steady loss of islet HS with increasing levels of islet inflammation (Figure 1D). In fact, HS loss appears to be a very sensitive marker of islet damage and subsequent development of Type I diabetes.

The next series of experiments used HPSE-1-deficient mice to identify the source of HPSE-1 in this model of acute T-cell-mediated inflammation. Initially, it was found that HPSE-1 deficiency in either the donor T cells or the RIP-OVA^hi^ recipient mice resulted in a small 2–3 day delay in diabetes onset. However, remarkably, the absence of HPSE-1 in **both** donor T cells and recipient mice resulted in a dramatic and highly significant increase in the frequency of diabetes-free mice from 0% in WT to ~60% in double KO mice (Figure 2B). Additional islet inflammation and HS staining data supported the conclusion that both donor and recipient HPSE-1 contribute to islet pathology (Figure 2C–E), with loss of islet HS again being a sensitive indicator of islet damage as early as 3 days after T cell adoptive transfer (Figure 2E). Subsequent studies with HPSE-1-deficient OT-II and OT-I T cells demonstrated, as expected, that antigen-activated CD4^+^ OT-II T cells are the dominant T cell population involved in diabetes induction (Figure 3B). Analysis of bone marrow chimeras, consisting of reconstituted HPSE-1 WT or HPSE-1-deficient RIP-OVA^hi^ host mice with either HPSE-1 WT or HPSE-1-deficient bone marrow cells, revealed that both radio-resistant hosts and haemopoietic-derived bone marrow cells contribute approximately equally to the diabetes incidence attributed to RIP-OVA^hi^ mice. Collectively, these data imply that HPSE-1 derived from at least three different cell types is required for maximum induction of Type I diabetes by a CD4^+^ T cell response. Obviously, one of these populations consists of the antigen-specific CD4^+^ T cells that initiate the islet inflammatory response. In fact, tracking of HPSE-1-deficient OT-I and OT-II T cells in the islets of HPSE-1-deficient hosts also revealed reduced numbers compared with their WT counterparts (Figure 4). The nature of the other two cell populations is less clear but we know, based on the bone marrow chimera data, that at least one is of haemopoietic origin and the other is radio-resistant. A very likely radio-resistant candidate is endothelial cells, particularly as it has been reported that these cells contribute to the local production of HPSE-1, impacting BM remodelling and leukocyte extravasation in a mouse model of delayed-type hypersensitivity [26]. On the other hand, the bone-marrow-derived source could be platelets as they constitutively carry very high levels of HPSE-1 that are released when platelets degranulate at sites of tissue injury and inflammation [27,28].

Finally, there is the intriguing observation that whenever RIP-OVA^hi^ mice are injected with naïve OT-1 and antigen-activated OT-II T cells that are HPSE-1 deficient there is always a subset of mice (~20–40%) that still become diabetic. It is remarkable that the deletion of the *Hpse-1* gene can have such a profound effect on CD4^+^ T cell effector function. Nevertheless, this result implies that there is an HPSE-1-independent mechanism of islet destruction occurring in this model although how it is mediated remains to be determined. Are we observing an invadopodia-driven entry of T cells into an inflammatory site or are there other ECM-degrading enzymes that can compensate for a lack of HPSE-1? This is clearly a research area that warrants further investigation.

## 4. Materials and Methods

### 4.1. Mice

HPSE-1-deficient C57BL/6J mice were kindly provided by Dr. I Vlodavsky and Dr. J Li [18]. C57BL/6J OT-I, OT-II, and RIP-OVA^hi^ mice [9,29] were obtained from the Walter and Eliza Hall Institute (WEHI), Australia. As the OT-II TCR is linked to the Y chromosome, only male donor and recipient mice were used. C57BL/6J mice normally express the CD45.2 allele of CD45 but a CD45.1 C57BL/6J strain has been developed that provides CD45.1 as a convenient marker for tracking injected leukocytes, with the Australian Phenomics Facility (ANU) breeding CD45.1 bearing strains of the ones listed above (and as indicated in the Figures). All mice were maintained in the specific-pathogen-free Australian Phenomics Facility (ANU). All animal procedures were approved by the Australian National University Animal Experimentation Ethics Committee (AEEC).

### 4.2. Diabetes Induction

Spleen and lymph nodes including inguinal, superficial cervical, and mesenteric LNs were dissected from male OT-I and OT-II mice. Single cell suspensions of OT-II T cells were made and stimulated with irradiated C57BL/6J splenocytes (220 rad) pulsed with 1 µg/mL ISQAVHAAHAEINEAGR (OVA_323–339_ peptide) in T75 flasks with 10 ng/mL rIL-2 and 5 µg/mL LPS, and with 0.5 µg/mL OVA_323–339_ peptide being also added to OT-II cultures [30]. Flasks were kept upright for 2 days and then laid flat for 2 days before harvest. A similar procedure was followed for OT-I T cells, but only for assessment of HPSE-1 expression following activation, using 1 µg/mL of SIINFEKL (OVA_257–264_ peptide). Peptides [31] were produced by the Biomolecular Resource Facility (JCSMR, ANU).

Naïve OT-I and activated OT-II T cells were purified by negative selection using hybridoma supernatants of M5/114, Ter119, M1/70, F4/80, and RB6.8C5. Additionally, GK1.5 mAb for OT-I T cells and 53.6.7 mAb for OT-II T cells (obtained from the WEHI mAb Facility) were used. This was followed by BioMag^®^ Goat anti-Rat IgG (Qiagen, Hilden, Germany), and a magnetic separator to deplete unwanted cells. Then, 200 µL of cell suspension (2x10^6^ OT-I and/or 2 × 10^6^ OT-II T cells) was injected i.v. into 3–5 male RIP-OVA^hi^ mice. Purity was checked by flow cytometry prior to transfer. Blood glucose was monitored daily for up to 14 days and mice were deemed diabetic and euthanised after 2 consecutive hyperglycaemic readings (>13 mmol/L). Diabetes incidence was plotted as ‘% Diabetes Free’ versus ‘Days Post Transfer’.

### 4.3. Bone Marrow Chimeras

8–10-week-old male WT and HPSE-1-deficient RIP-OVA^hi^ (CD45.2) mice were irradiated with 2 doses of 450 rad and were reconstituted with 5 × 10^6^ wild type or HPSE-1-deficient bone marrow cells (CD45.1) [9,19]. After 8 weeks of reconstitution, chimeras were i.v. injected with *Hpse-1* knockout OT-I and OT-II T cells as described above and were monitored for diabetes development.

### 4.4. Flow Cytometry

Methods of Carboxyfluorescein succinimidyl ester (CFSE) and Cell Trace Violet (CTV) (Molecular Probes, Eugene, OR, USA) labelling to monitor OT-I and OT-II T cell migration in vivo were performed as published [9,21,32,33]. Cells were labelled before injection, and recipient mice were euthanised at different time points after transfer for flow cytometric analysis. Splenocytes and LN cells were obtained as above. Islets were isolated with a standard collagenase protocol [13]. Isolated islets containing infiltrated lymphocytes were cultured with RPMI1640/10%FCS overnight in a 37 °C incubator to enable CD4 and CD8 T cell egress and recovery. Lymphocyte subsets and macrophages were identified using fluorescent antibodies for CD4, CD8, CD11b, CD45.1, CD45.2, F4/80, TCR Vα2, and 7AAD (BD). Heparanase-1 was detected with the Hpa 3/17 mAb supplied by Abcam, Cambridge, UK. Data were acquired on a BD LSR II and analysed using FlowJo software (Treestar, Becton Dickinson, Franklin Lakes, NJ, USA).

### 4.5. Pancreatic Islet Histology, Inflammation, and HS Staining

Pancreata were fixed in neutral-buffered formalin and 4 µm paraffin sections were stained for H&E and Alcian blue [13,17]. All islets from 4 sections (at 76-micron intervals) were counted for semi-quantitative analysis of islet infiltration and islet HS. Similarly, images of all islets in an additional 4 sections (at 76-micron intervals) were analysed by ImageJ software (NIH) to quantify the Alcian-blue-positive area within islets as an indicator of HS content [13].

### 4.6. Statistical Analysis

Prism software (GraphPad Software, San Diego, CA, USA) was used to perform statistical tests and to generate graphs. Statistically significant differences between groups were obtained by Fisher Exact Test (Figure 2B and Figure 3B,D), by Student’s *t*-test (Figure 1D, Figure 2D, and Figure 4B), by Mann–Whitney test (Figure 2E) and by the non-parametric Kruskal–Wallis test with Dunn’s multiple comparisons test (Figure 4D). Data expressed as mean ± s.e.m. * *p* < 0.05, ** *p* < 0.01, *** *p* < 0.001, **** *p* < 0.0001.

## Figures and Tables

**Figure 1 ijms-23-04625-f001:**
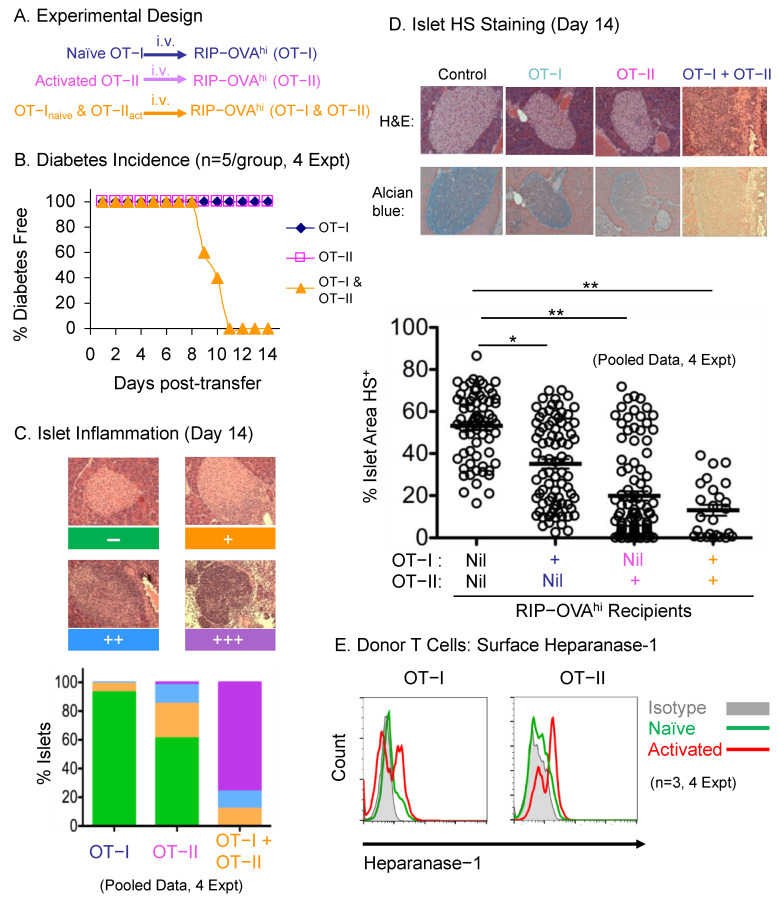
Antigen- specific T cell transfer model for studying islet infiltration and HS degradation (**A**) Experimental design. Adoptive transfer of naïve OT-I cells and in vitro activated OT-II cells into RIP-OVAhi mice to induce diabetes. (**B**) Frequency of diabetes free RIP-OVAhi mice in the 3 groups (5 mice/group) of recipients receiving naïve OT-I cells (blue), activated OT-II cells (purple), or both OT-I and OT-II (orange) T cells. (**C**) Intensity of islet inflammation in the pancreas at day 14. Islet infiltration was classified into 4 grades: intact islet, <50% of the islet infiltrated, >50% of the islet infiltrated, and 100% of the islet destroyed. Data are compiled from 4 independent experiments. (**D**) Representative pancreas histology at day 14 stained with H&E or Alcian blue from RIP-OVAhi mice transferred with OT-I or OT-II T cells. Quantification of HS+ area stained with Alcian blue of individual islets in 4 sections of each recipient pancreas. Each dot represents the value of one individual islet (5 mice per group). Significance was assessed by Student’s *t*-test, * *p* < 0.05, ** *p* < 0.01. (**E**) Histogram overlay of HPSE-1 staining in OT-I and OT-II T cells on the cell surface before and after OVA stimulation in vitro (Grey infill: isotype control; green line: naïve; red line: activated).

**Figure 2 ijms-23-04625-f002:**
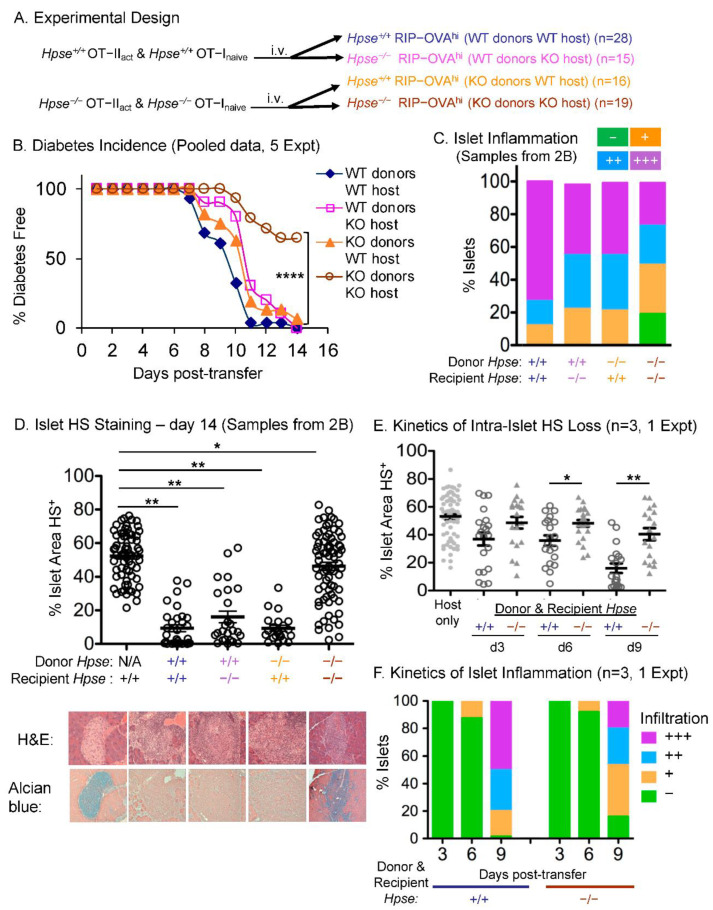
HPSE-1 drives antigen-specific T cell induction of T1D. (**A**) Experimental design. HPSE-1 WT or deficient naïve OT-I and activated OT-II cells were each injected into two groups of HPSE-1 WT and deficient RIP-OVAhi mice. (**B**) Diabetes incidence. WT donors and hosts (blue n = 28), WT donors and HPSE-1 deficient hosts (purple n = 15), HPSE-1 deficient donors and WT hosts (orange n = 16), deficient donors and hosts (brown n = 19). Results were pooled from 5 experiments. Significance was assessed by Fisher’s exact tests. **** *p* < 0.0001. (**C**) Intensity of islet inflammation on day 14 post-transfer using samples from 2B. (**D**) Content of intra-islet HS. HS+ area stained with Alcian blue of individual islets in 4 sections of each recipient mouse pancreas, 5 mice per group, one representative experiment. Each dot represents the value of one individual islet. Significance was assessed by Student’s *t*-test, * *p* < 0.05, ** *p* < 0.01. (**E**) Kinetics of intra-islet HS loss. Graph shows “% islet area HS+” of individual islets in 2 sections of each pancreas, 6 mice per group, 3, 6 and 9 days after transfer. Data are pooled from two separate experiments. Each point represents one individual islet. Significance was assessed by Mann-Whitney test, * *p* < 0.05, ** *p* < 0.01. (**F**) Kinetics of islet inflammation. Data show the average of 6 mice from 2 experiments at each time point.

**Figure 3 ijms-23-04625-f003:**
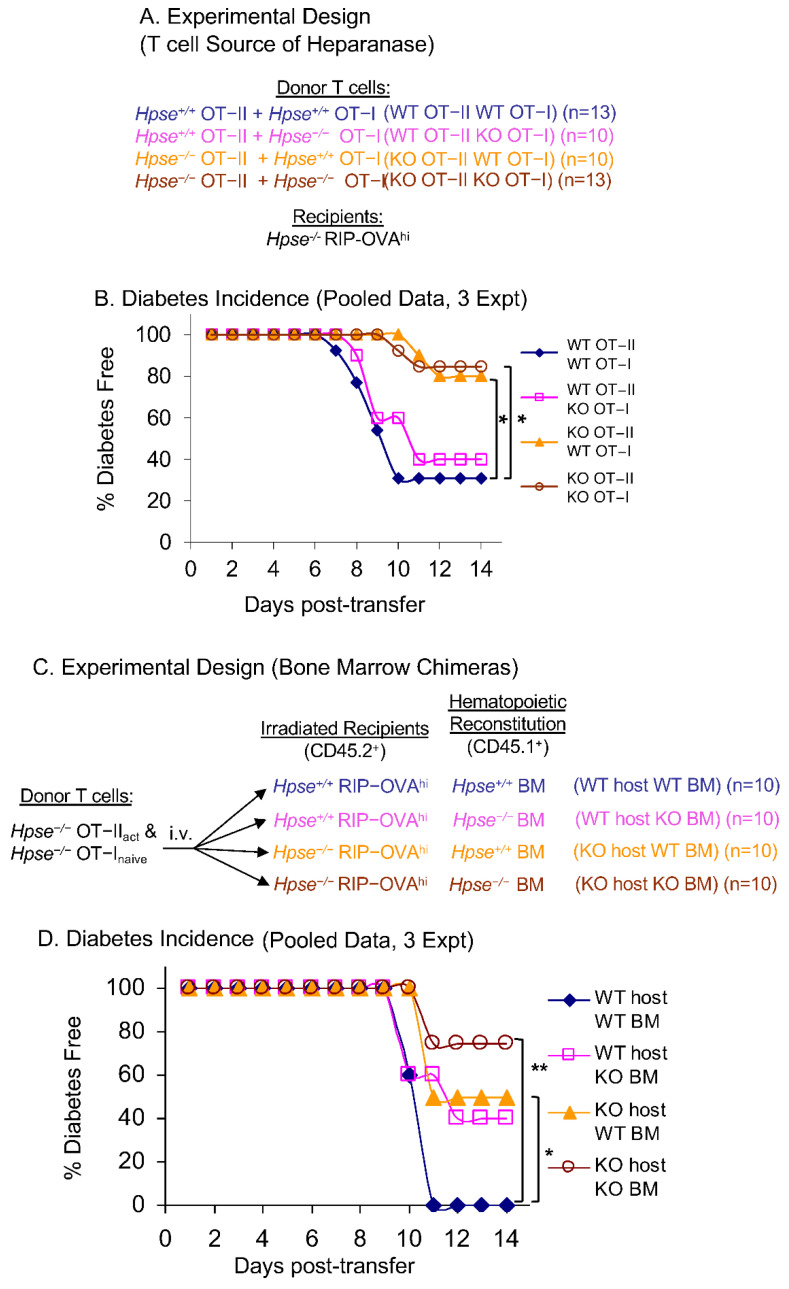
HPSE-1 in OT-II CD4^+^ T cells plays a key role in diabetes initiation but HPSE-1 from hematopoietic and radiation-resistant host cells also contribute to lymphocyte infiltration. (**A**) Experimental design. HPSE-1 WT or deficient naïve OT-I T cells were mixed with WT or HPSE-1 deficient activated OT-II T cells and transferred to HPSE-1 deficient RIP-OVAhi recipients. (**B**) Diabetes incidence showing frequency of diabetes free recipients. WT OT-II and OT-I (blue n = 13), WT OT-II with deficient OT-I (purple n = 10), deficient OT-II with WT OT-I (orange n = 10), and deficient OT-II and OT-I (brown n = 13). Results were pooled from 3 experiments. Significance was assessed by Fisher’s exact test *: *p* < 0.05. (**C**) Experimental design. HPSE-1 WT or deficient RIP-OVAhi recipients (CD45.2) were irradiated, reconstituted with either HPSE-1 WT or deficient bone marrow (CD45.1), and transferred with HPSE-1 deficient OT-I and OT-II cells (CD45.1^+^CD45.2^+^) after full reconstitution. (**D**) Diabetes incidence. Data are compiled from 2 independent experiments, 10 mice/group. Significance was assessed by Fisher’s exact test *: *p* < 0.05, ** *p* < 0.01.

**Figure 4 ijms-23-04625-f004:**
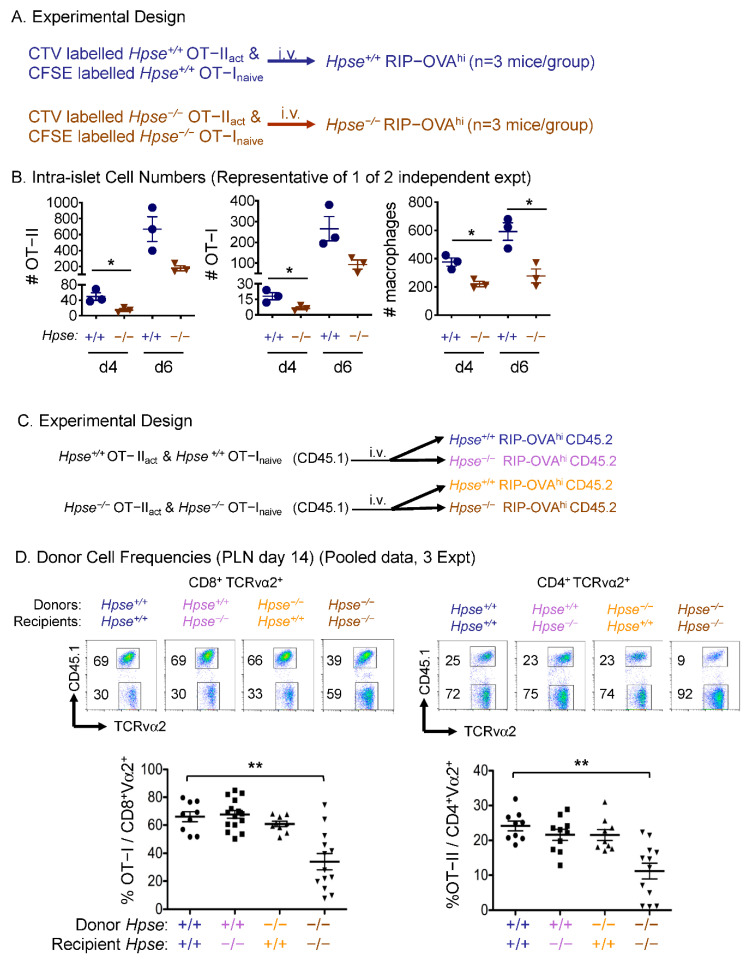
Cellular tracking of transferred OT-I and OT-II T cells in the presence or absence of HPSE-1. (**A**). Experimental design. WT or HPSE-1 deficient activated OT-II and naïve OT-I cells were, respectively, fluorescently labelled with CTV or CFSE and transferred into HPSE-1 genotype matched RIP-OVAhi hosts. (**B**) Intra-islet donor OT-I, OT-II and host CD11b+ F4/80+ (macrophage) numbers per pancreas on day 4 and 6 after transfer. Each point represents cell counts from one individual mouse, WT (blue), deficient (purple). Data are representative of 1 of 2 independent experiments. Significance was assessed by Student’s *t*-tests, * *p* < 0.05. (**C**) Experimental design. HPSE-1 WT or deficient CD45.1 naïve OT-I and activated OT-II cells were each injected into two groups of CD45.2 HPSE-1 WT and deficient RIP-OVAhi mice. (**D**) Upper graphs: Representative flow cytometry plots of different treatment groups, with appropriately gated populations. Lower graphs: Frequency of donor OT-I and OT-II cells in the PLNs of individual recipients at day 14. Data are pooled from three independent experiments. Significance was assessed by the non-parametric Kruskal–Wallis test with Dunn’s multiple comparisons test, ** *p* < 0.01.

## Data Availability

Data reported in this study is stored at the JCSMR and is freely available to other researchers upon request.

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
