# Peer review of "Acute T-Cell-Driven Inflammation Requires the Endoglycosidase Heparanase-1 from Multiple Cell Types"

_ijms, 2022, doi:10.3390/ijms23094625_

Round 1

Reviewer 1 Report

In this manuscript (ijms-1657694), Wu et al examined the significance of heparanase in the development of type 1 diabetes. More specifically, the authors focus on the involvement of heparanase in CD4 and CD8 T cells-driven inflammation of the pancreas in eliciting diabetes. This notion emerges from previous reports of this group, nicely showing that HS function to protect the insulin-producing beta cells, and cleavage of HS by heparanase-positive inflammatory cells impedes this protective effect, leading to the onset of diabetes. In the current research, the authors elegantly use sophisticated mouse models, genetic manipulations, and adoptive transfer technologies, to examine the significance of heparanase contributed by the host or the implanted CD4 T cells (OT-II) or CD8 T cells (OT-I) in the onset of diabetes. The manuscript is well written and well organized and the overall quality of the data and its presentation is impressive. The data appears very strong and clearly show that heparanase contributed by inflammatory T cells and macrophages plays a critical role in the onset of diabetes in this mouse model. I only have minor suggestions and encourage the publication of this study.

Minor points

  1. In Figure 2E, statistics are missing. Please add.
  2. In Figure 4B, analyses are performed on day 4 and not on day 3 as in the other figures. Please explain the rationale.
  3. In Figure 4B, left and middle panels, there seems to be a statistically significant difference between heparanase positive (+/+) and heparanase negative (-/-) cells, but the scale makes it difficult to know the actual numbers. Please consider changing the scale to the smaller values and then jump to the big ones.
  4. It seems (figure 4B) that macrophages are being recruited to the inflamed pancreas, likely by mediators/ chemo-attractants secreted by the T cells. Can the authors speculate on possible T cells-derived cytokines and chemokines responsible for the recruitment of macrophages?
  5. In Figure 4B, the authors quantify the number of T cells in PLN. It is unclear to this reviewer why the analyses were not performed on pancreas tissue or isolated islets. Please explain the rationale.
  6. In lines 336-337, one 'Then' is avoidable.

Author Response

Please see attached Word document: Author's reply to the Review Report (Reviewer 1) Final

Reviewer 2 Report

Wu et al described the role of HPSE1 in a model of T cell driven inflammation. They showed that HPSE1 is required to obtain fully aggressive type 1 diabetes. HPSE1 expression is required in antigen specific CD4+ T cells transferred but also in the host in both hematopoietic and non-hematopoietic cells. The paper is well designed with elegant mouse models but several minor points need to be addressed:

-The authors should clarify in each experiments the number of independent experiments and the number of animal. In particular for the experiment 4B where a limited number of experiments seems to be the cause of a lack of significance.

-Figure 1D: the control and the OT-I + OT-II groups should be shown.

-Lane 200-201: “provide evidence for both hematopoietic AND radiation resistant cells”

-Lane 202: “HPSE-1 that degrade HS to enable HPSE-1 deficient OT-I and OT-II cells to enter islets and induce diabetes” not shown at this point.

-Lane 213: “Thus, delayed initiation of infiltration by HPSE-1 deficient antigen-specific T cells at early time points”. Rather inhibited than delayed since infiltration is still lower at day 14.

-Lane 348: The reference of the ant-HPSE1 ab for flow cytometry must be specified.

-The hypothesis of the role of platelets and endothelial cells in HPSE1 dependent inflammation could be mentioned in the discussion but should be mitigated in the summary since speculative.
